# Influence of Post Processing on Thermal Conductivity of ITO Thin Films

**DOI:** 10.3390/ma16010362

**Published:** 2022-12-30

**Authors:** Anna Kaźmierczak-Bałata, Jerzy Bodzenta, Mohsen Dehbashi, Jeyanthinath Mayandi, Vishnukanthan Venkatachalapathy

**Affiliations:** 1Institute of Physics, Silesian University of Technology, Konarskiego 22B, 44-100 Gliwice, Poland; 2Department of Physics, University of Oslo, Blindern, P.O. Box 1048, NO-0316 Oslo, Norway; 3Department of Materials Science, School of Chemistry, Madurai Kamaraj University, Madurai 625021, India; 4Department of Materials Science, National Research Nuclear University “MEPhI”, 31, Kashirskoesh, 115409 Moscow, Russia

**Keywords:** ITO thin films, post processing, thermal conductivity, transparent conductive electrodes

## Abstract

This work presents the influence of post processing on morphology, thermal and electrical properties of indium tin oxide (ITO) thin films annealed at 400 °C in different atmospheres. The commercially available 170 nm thick ITO layers deposited on glass were used as a starting material. The X-ray diffraction measurements revealed polycrystalline structure with dominant signal from (222) plane for all samples. The annealing reduces the intensity of this peak and causes increase of (221) and (440) peaks. Atomic force microscopy images showed that the surface morphology is typical for polycrystalline layers with roughness not exceeding few nm. Annealing in the oxygen and the nitrogen-hydrogen mixture (NHM) changes shapes of grains. The electrical conductivity decreases after annealing except the one of layer annealed in NHM. Thermal conductivities of annealed ITO thin films were in range from 6.4 to 10.6 W·m^−1^·K^−1^, and they were higher than the one for starting material—5.1 W·m^−1^·K^−1^. Present work showed that annealing can be used to modify properties of ITO layers to make them useful for specific applications e.g., in ITO based solar cells.

## 1. Introduction

Indium tin oxide represents a group of wide band gap semiconductors, which exhibits high electrical conductivity and high optical transparency. The ITO has the energy band gap greater that 3.2 eV, high transmittance in visible and near IR regions, greater that 70%, and high electrical conductivity, greater than 10^3^ S·cm^−1^. Typical measured values of the energy band gap are in range (3.3–4.3) eV, and the sheet resistance equals to 20 Ω/□ [1]. Therefore, it fulfils requirements for transparent conducting electrodes (TCO)—the transparency over visible light region and effective carrier transport. The ITO thin films deposited on glass are widely applied as transparent conductive electrodes in solar cells, flat panel displays, organic light emitting diodes, transistors and defoggers [2,3,4,5,6,7].

ITO is formed by doping the indium oxide with tin (Sn), which substitutes the In^3+^ ions in the cubic bixbyite structure of In_2_O_3_. The tin and oxygen vacancies play a dominant role in carrier transport in the lattice [8]. Substitution of In^3+^ with Sn^4+^ results in addition of free electrons to the lattice and improvement of electrical conductivity. The structure and physical properties depend on fabrication process and growth conditions like oxygen partial pressure [9]. It was shown that the ITO thin films transmittance increased from 93% to 99.6% with the increase in oxygen partial pressure ratio from 0 to 27%. The carrier’s concentration increased from 5.23 × 10^18^ cm^−3^ to 4.54 × 10^20^ cm^−3^ and the carrier mobility varied in range (13.1–14.6) cm^2^·V^−1^·s^−1^ with the reduction in this ratio from 16% to 2%. ITO layers are produced by various techniques, e.g., direct current (DC) and radio-frequency (RF) magnetron sputtering, pulsed laser deposition [10], vacuum thermal evaporation [11], dip-coating wet technology [12] and oblique angle deposition [13]. Post-deposition annealing is often applied to promote crystallinity of the layers and enhances the electrical conductivity and optical transmittance of ITO films [14]. The rapid thermal annealing in temperature range (520–670) °C for 10 min. was proposed to fabricate high quality ITO thin films [15]. Investigations of annealing temperature influence in range (150–550) °C on properties of ITO thin films was shown in [16]. The ITO layers have been fabricated by the sol-gel process and post annealed in various temperatures. The crystallinity, optical and electrical properties were improved, what was especially pronounced for temperature equal to 550 °C. Physical properties such as the thermal conductivity, the free carrier density, the carriers mobility, and optical properties are crucial parameters determining application of ITO layers. These parameters are sensitive to microstructure and quality of the layer. The thermal conductivity depends on morphology and thickness of thin film and typically decreases with decrease in layer thickness, what is especially noticeable for thicknesses below 1 μm [17,18,19,20]. This fact contributes to difficulties of theoretical predictions of thermal properties of nano-scale layers. Thermal properties of thin films can be modified by doping and by changes of crystalline structure, which depend on fabrication and post processing parameters. Thus, understanding of thermal transport in thin films is important for engineering thermal properties for selected applications.

In this work we report on results from structural, thermal and electrical properties investigations of annealed ITO thin films deposited on glass substrates. The annealing in various atmospheres resulted in the increase in the thermal conductivity value and changes in surface morphology of layers. The post processing can be applied as an alternative method to modify layers properties towards particular demand.

## 2. Materials and Methods

In this work we investigated a set of commercial ITO thin films (Hoya, Japan) deposited on glass substrate. The thickness of films used in the present work was 170 nm. The ITO films were annealed at 400 °C for 1 h in various atmospheres: air (sample S2), Nitrogen-Hydrogen mixture (NHM, sample S3), oxygen, nominal purity of 97% (O_2_, sample S4)) and nitrogen (N_2_, sample S5). The heating and cooling rates were about 3 °C/min. The as-received sample S1 was used as a reference. The surface topography of ITO samples was investigated by AFM. Topographical scans were recorded using the XE-70 model (Park Systems Inc., Suwon, Republic of Korea) working in a contact mode. The Budget Sensors ContAl-G probes with Al reflecting coating (resonant frequency 13 kHz, force constant 0.2 N/m) were used. For quantitative analysis, i.e., surface roughness *R_a_* and grains’ size distribution, the built-in Gwyddion algorithms were used. The structure of ITO thin films was studied by analysis of X-ray diffraction (XRD) technique. The XRD spectra were recorded on Rigaku Ultima II Max using Cu Kα radiation. The thermal conductivity *k* of ITO layers was determined applying SThM method. Measurements were conducted on the same XE-70 scanning probe microscope with SThM module. The KNT-SThM-1a thermal probe (Kelvin NanoTechnology, Glasgow, UK) was used in measurements. The probe was working in active mode simultaneously heating a sample and measuring its temperature. In this mode the probe was driven by DC current equal to 1.9 mA with a small AC component of 0.1 mA amplitude superposed on it. The static and dynamic electrical resistances of the probe were measured as the SThM signals applying lock-in detection (SR-830 DSP lock-in amplifier, Stanford Research Systems, Sunnyvale, CA, USA). The SThM quantitative measurements require calibration of each thermal probe before measurements. The probe calibration was based on SThM signal measurement carried out for a set of reference samples of known thermal conductivities. The thermal conductivity of thin films was determined from calibration curve. Several examples of thin film thermal conductivity measurements can be found in [21,22,23]. The method allows investigation of thermal properties with a high spatial resolution [24,25,26,27,28]. The modification of SThM method based on measurements of static and dynamic resistance of thermal probe for quantitative thermal measurements was proposed in [29]. The sheet resistance measurements were performed using a four-point probe (Keithley 4200-SCS, Tektronix, Inc., Beaverton, OR, USA). The electrical conductivity of ITO layers was calculated as the inversion of the sheet resistance and the film thickness product. The electrical resistance, the Hall coefficient, carriers concentration and the mobility in room temperature were determined by applying the Van der Pauw method (LakeShore 7604, Lake Shore Cryotronic, Inc., Westerville, OH, USA). The Hall measurements in the van der Pauw configuration enable determination of the electrical properties of samples in a wide range of magnetic fields up to 1.3 T and temperatures.

## 3. Results and Discussion

### 3.1. Structural Study

The structure of ITO was studied by analysis of XRD spectra recorded under conventional (0–2θ) scanning configuration. The ITO thin films show the bixbyite type cubic lattice structure of space group Ia3 (206). The lattice constant of an undoped In_2_O_3_ structure is equal to 10.1195 Å, while substitution of In^3+^ with Sn^4+^ ions result in the increase in a to 10.1517 Å. The characteristic reflections from the following planes: (211), (222), (332), (431), (440), (433), (444) with corresponding peaks positions for In_2_O_3_ and (300) plane for In_4_Sn_3_O_12_ components in the XRD pattern are depicted in Figure 1a [30,31]. The reflection from (222) plane corresponding to In_2_O_3_ component was dominant for all samples, however the fraction of In_4_Sn_3_O_12_ was also noticeable, as shown in Figure 1b. The decrease in the intensity of characteristic reflection from plane (222), compared to as-received sample, could result from more compacted layers or possible degradation of the film quality after annealing at 400 °C in air, NHM, O_2_ and N_2_ atmosphere.

The mean crystallite size *Λ* was estimated from the Debye-Scherrer’s formula [32]:(1)$$\Lambda =\frac{\alpha \lambda }{\beta cos\theta },$$
where *α* is shape factor, *λ* is the X-rays wavelength, *θ* is the Bragg diffraction angle and *β* is the broadening of the diffraction peak measured at half of its maximum intensity (FWHM). The shape factor value *α* = 0.9 was assumed in the calculations. The *Λ* parameter varied in range (60–66) nm. The analysis of the XRD signal amplitude for all identified peaks shows that, except for one reflection, all signals come from the In_2_O_3_ compound. Detailed analysis of spectra revealed that the (222) reflection peak is composed of two components, In_2_O_3_ (222) and In_4_Sn_3_O_12_ (300) peaks. (Figure 1b). The *β* parameter of In_4_Sn_3_O_12_ peak is the largest among all peaks, which may indicate low crystallinity of this compound in the form of nanocrystallites woven into In_2_O_3_ grains.

The amplitudes of all identified reflection peaks corresponding to In_2_O_3_ and In_4_Sn_3_O_12_ compounds basing on XRD spectra analysis are gathered in Table 1. The quantitative analysis of amplitudes of the diffraction peaks shows that annealing in different atmospheres resulted in a decrease in the signal for the dominant (222) reflection, which may indicate a deterioration of crystallinity of the layers. However, in the case of (300) reflection for In_4_Sn_3_O_12_ compound, the signal amplitude increased for all samples except for sample S4 annealed in O_2_. The amplitude of XRD spectra for ITO layers is depicted in Figure 1c.

### 3.2. AFM Investigations

In order to quantitatively analyze the surface of the ITO layers, the following parameters were determined from AFM topographical images: surface roughness *R_a_* (arithmetic mean of vertical deviations from the mean value), and the grain size distribution. The lowest *R_a_* equal to 1.194 nm was recorded for a sample S4, annealed in O_2_ atmosphere. The largest value of *R_a_* was determined for as-received sample S1 and the one annealed in air S2, the *R_a_* values were 2.013 nm and 2.044 nm, respectively. The roughness of the sample S3 annealed in NHM atmosphere was similar to the one annealed in oxygen and equaled to 1.396 nm. The samples S1 is composed of elongated grains with sizes in between 8 nm and 20 nm. The mean grain size is 12.8 nm, and the grain aspect ratio is of about 2.7. The sample S2 mainly consists of densely packed, needle-like grains with sizes similar to the ones of sample S1. The mean grain size is 12.6 nm, and the grain aspect ratio is ~2.9. Some texturization can be observed in topography images. Samples S3 and S4 are composed of smaller grains with sizes ranging from 6 nm to 12 nm. The sample S4 has also a few big clusters of ~20 nm size. Grains are elongated with aspect ratio ~1.6 and ~2.0 for samples S3 and S4, respectively. The results correlate with *R_a_* parameters, which are similar for samples S1 and S2 (~2.0 nm), and S3 and S4 (~1.3 nm). Generally, the ITO thin films are composed of small, elongated grains uniformly distributed in the layer with the mean surface roughness not exceeding few nm. Topographic images and mean grain size distribution of ITO layers are gathered in Figure 2.

### 3.3. Thermal and Electrical Properties

The static (*R_s_*) and the dynamic (*R_d_*) thermal resistances were measured for the probe in contact with the sample and lifted 2.0 mm over the sample surface. Detailed description of the measuring procedure can be found in Bodzenta, et al. [29]. The thermal conductivity of the sample can be related to the ratio of *R_d_*−*R_s_* differences determined for the probe in contact with the sample and the probe lifted
(2)$$S=\frac{{\left(R_{d}-R_{s}\right)|}_{contact}}{{\left(R_{d}-R_{s}\right)|}_{air}}$$

This ratio is hereinafter referred to as SThM signal. The SThM signals obtained for four reference samples (PET, fused quartz, glassy carbon, and YAG) and ITO samples (as received, annealed in air, NHM, oxygen) are collected in Table 2. ITO sample annealed in N_2_ was not available for SThM measurements. The thermal conductivities of reference samples and the ones of ITO samples determined from a calibration curve are also shown. The calibration curve was obtained by interpolation of SThM signal obtained for reference samples, what was shown in Figure 3. The thermal conductivities of ITO samples shown in Table 2 are the ones obtained from calibration curve. These are apparent thermal conductivities of the thin layer—substrate system. The apparent thermal conductivity of the system *k_s_* is correlated with the spreading resistance *R_s_* in Equation (3)
(3)$$R_{s}=\frac{1}{4ak_{s}}$$

The effective spreading resistance *R_s_* of the system built of a thin layer on a thick substrate is given by the formula
(4)$$R_{s}=\frac{1}{\pi dk_{2}}\frac{x}{y}\int_{0}^{\infty }\frac{\left(y+1\right)exp\left(\zeta x\right)+\left(y-1\right)exp\left(-\zeta x\right)}{\left(y+1\right)exp\left(\zeta x\right)-\left(y-1\right)exp\left(-\zeta x\right)}\cdot \frac{J_{1}\left(\zeta \right)sin\zeta }{\zeta^{2}}d\zeta ,$$
where *d* is the film thickness, *k*_2_ is the substrate thermal conductivity, *x* = *d*/*a*, *y* = *k*_1_/*k*_2_, and ζ is the dummy variable of integration [33]. As the layer thickness *d* = 170 nm, the contact radius *a* = 100 nm, and the thermal conductivity of the substrate *k*_2_ = 1.1 W·m^−1^·K^−1^ are known, the real thermal conductivity of the layer k_1_ can be determined from theoretical curve, represented by two dimensionless variables: thermal conductivity ratio y and layer thickness to thermal probe radius ratio *x* (Equation (4)). Assuming known thermal conductivity of the substrate *k*_2_, thickness of the layer *d* and thermal conductivity *k_s_* for ITO/Glass sample determined from SThM measurements it was possible to calculate the corrected thermal conductivity of ITO thin film as *k*_1_ = *y*·*k*_2_. The SThM signal measurement uncertainties were in range 1–2%, calculated as a standard uncertainty based on the number of repeated measurements. The thermal conductivity of thin films was determined from probe calibration measurements and *k* uncertainties were in the range 15–20%.

The electrical conductivity of reference ITO thin film S1 is 5.06 kS·cm^−1^. Layers annealed in air and O_2_, (S2 and S4) show the electrical conductivities equal to 2.79 and 2.11 kS·cm^−1^, respectively. These are the lowest electrical conductivities. The highest electrical conductivity was obtained for NHM annealed sample S3—5.53 kS·cm^−1^. This results from the highest carrier’s concentration and relatively high carries mobility. The electrical conductivity, carriers’ concentration and mobility for ITO samples are gathered in Table 3 and depicted in Figure 4a.

Taking into account uncertainties of measurements the thermal conductivities of samples S3, and S4 do not differ significantly. They are ~20% higher than the one of as-received sample S1. The highest thermal conductivity was obtained for sample S2. The thermal conductivities, phonon and electron component of the total thermal conductivities for ITO samples are depicted in Figure 4b for clarity.

The thermal conductivity connected with carriers *k_el_* is defined in terms of the electrical conductivity *σ* by the Wiedemann-Franz law, stating that carrier contribution of the thermal conductivity *k_el_* to the electrical conductivity *σ* of metals is directly proportional to the temperature *T*:
*k_el_*/*σ* = *L·T*
(5)

where *L* = 2.44·10^−8^ V^2^·K^−2^ is the Lorentz number.

The thermal conductivity carried by phonons was estimated by subtracting *k_el_* from the total thermal conductivity of thin films determined from SThM measurements. The results for the corrected thermal conductivity, and the phonon and electron components were gathered in the Table 4. 

The thermal treatment increased the thermal conductivity value for all samples. The thermal conductivity carried by phonons was constant for samples annealed in NHM and O_2_, what could suggest that the phonon mean free path should be almost constant for two ITO thin films. This observation is corroborated by topography images of sample S3 and S4. The layers are composed of rounded, densely packed grains and the surface roughness is about 1.3 nm. The phonon component of the thermal conductivity of ITO layer annealed in air (S2) is three times larger in comparison to sample S3 and S4, pointing to phonon heat transport in the thin film. Relatively high thermal conductivity can result from effective phonon transport along needle-like grains. The different trend is also reflected in the topography image. The layer consists of densely packed, needle-like grains arranged in sections resembling texturization. The surface roughness equal to 2 nm is also larger for sample S2.

The differences in surface morphology of ITO layers are not fully consistent with parameters determined from XRD spectra analysis. The mean grain size of ITO layers is equal to around 63 nm. This observation could suggest that thermal treatment resulted more in changes of grains orientation and less in changes of mean size of grains. The thermal conductivity carried by electrons is higher in comparison to the phonon part for as-received sample and samples annealed in NHM and oxygen atmosphere. The contribution of *k_el_* is about 60% of the total thermal conductivity values.

## 4. Conclusions

In this work the effect of annealing atmosphere on thermal, electrical, and structural properties of indium tin oxide thin films was investigated. The preferential crystal orientation along (222) plane was confirmed for all samples by the XRD spectra analysis. The post processing resulted in slight decrease of (222) peak intensity for ITO layers annealed at 400 °C. The AFM images proved uniformity of layers with surface roughness parameter *R_a_* not exceeding 2.1 nm. The layer annealed in air was composed of densely packed, elongated grains with some texturization visible. The grains orientation could suggest anisotropy of thermal properties of layer with relatively high conductivity along grain axis. The layers annealed in NHM, and oxygen were built from more rounded grains and were characterized by smoother surface, the value of *R_a_* parameter equalled to 1.4 nm and 1.2 nm, respectively. The post processing increases their thermal conductivities The thermal conductivity of ITO thin films annealed in NHM, and oxygen atmosphere equalled to 6.7 and 6.4 W·m^−1^·K^−1^, what was the increase of 21–24% compared to 5.1 W·m^−1^·K^−1^ for as-received sample. The highest thermal conductivity value equal to 10.6 W·m^−1^·K^−1^ was determined for sample annealed in air, what was the doubled value of the thermal conductivity in comparison to as-received ITO. Surprisingly, this sample has the lowest electrical conductivity. A possible explanation is that sample S2 is built of well oriented, tightly packed, elongated grains. It may result in anisotropy in the thermal conductivity with relatively high phonon conductivity along grains. High phonon thermal conductivity means low phonon scattering on defects. However, defects can create levels in the energy gap which can lead an increase in carrier’s concentration. Therefore, low density of defects can result in lower electrical conductivity. This is a hypothesis which should be verified experimentally.

Obtained results show that annealing in selected atmosphere can be an effective tool for controlling electrical and thermal transport in ITO layers. Therefore, post processing allows controlled modification of as-received layer properties and their adjustment for specific application.

## Figures and Tables

**Figure 1 materials-16-00362-f001:**
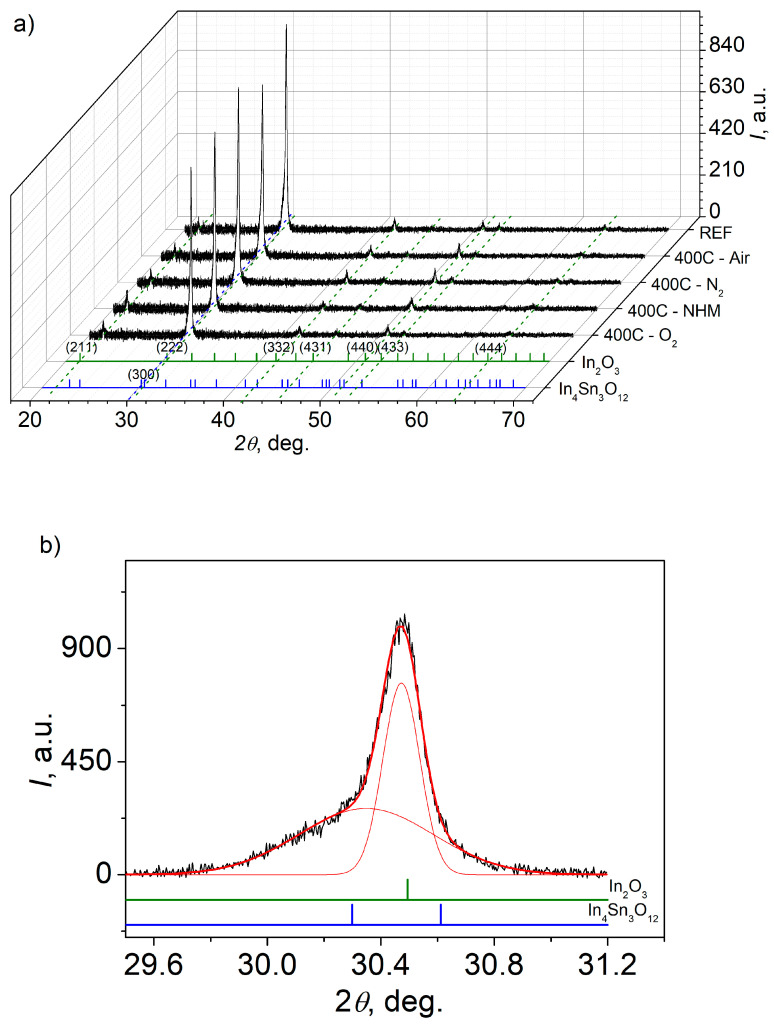
XRD spectra recorded for ITO thin films: as-received (S1) and annealed at 400 °C in various atmospheres (S2–S5) (**a**), exemplary diffraction peak analysis from plane (222) for as-received sample (**b**), amplitudes of reflection signals for ITO layers (**c**).

**Figure 2 materials-16-00362-f002:**
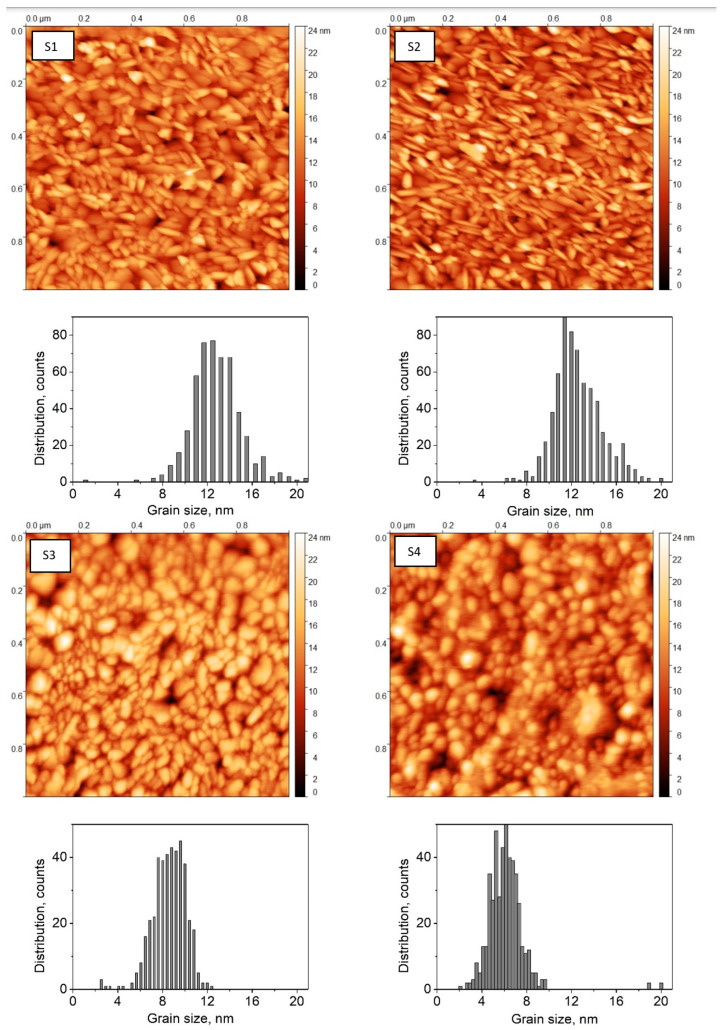
AFM topographic images of ITO layers and mean grain size distribution graphs.

**Figure 3 materials-16-00362-f003:**
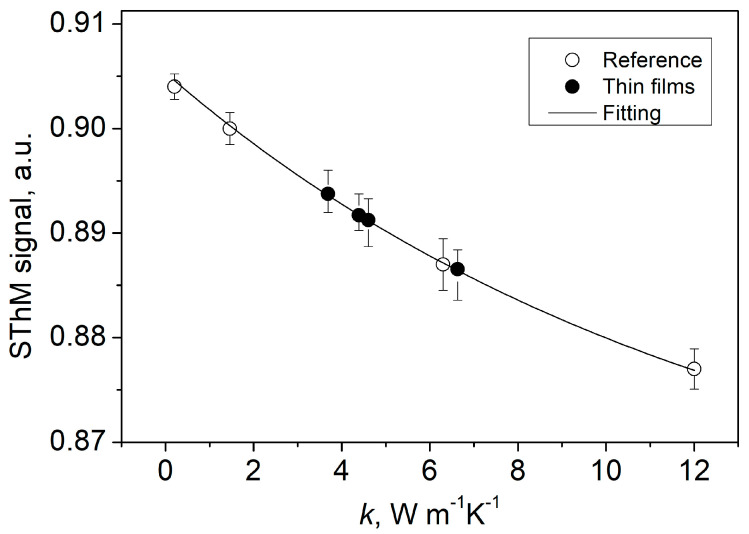
SThM signal vs. the thermal conductivity for reference samples (white circles) fitted with exponential curve (solid line). Points for ITO thin films (black circles) are placed on reference curve.

**Figure 4 materials-16-00362-f004:**
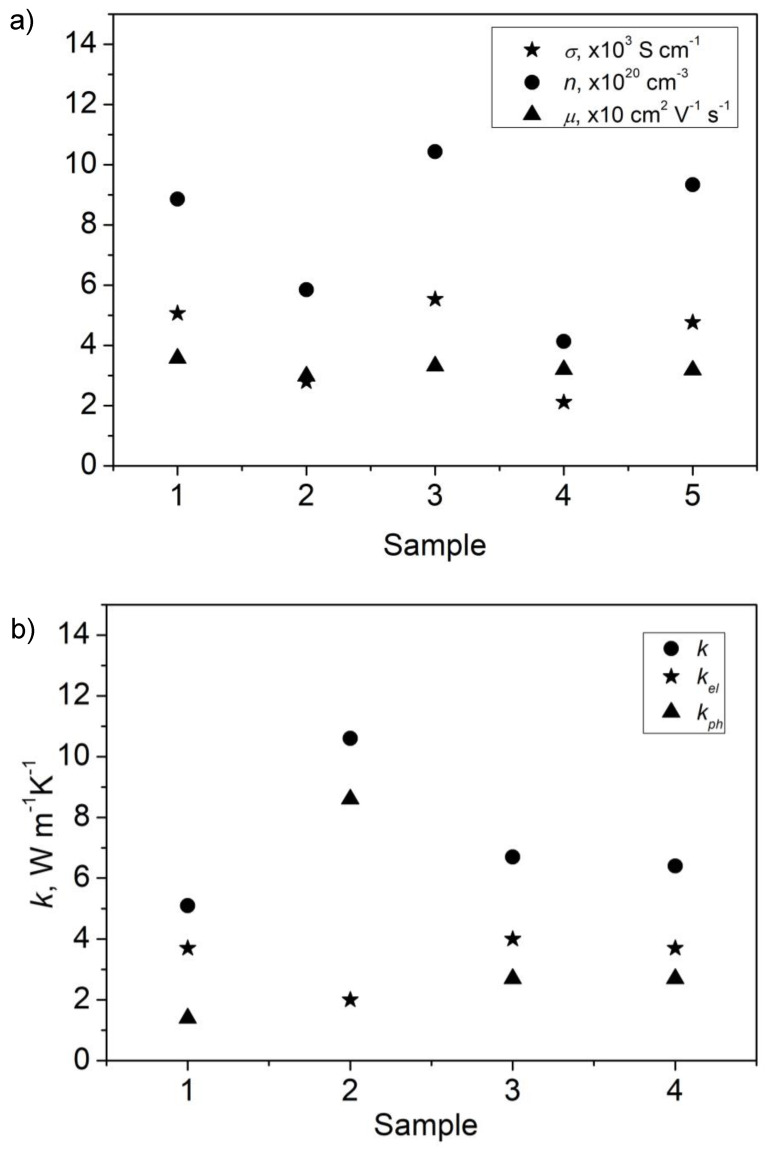
Electrical conductivity, carriers’ concentration and mobility of ITO thin films (**a**), and thermal conductivities, with electron and phonon components shown (**b**).

**Table 1 materials-16-00362-t001:** Amplitudes of pronounced reflection peaks corresponding to In_2_O_3_ and In_4_Sn_3_O_12_ compounds.

	S1	S2	S3	S4	S5
(221) In_2_O_3_	29	40	63	47	50
(300) In_4_Sn_3_O_12_	263	283	322	205	320
(222) In_2_O_3_	763	565	517	606	658
(332) In_2_O_3_	44	39	23	31	43
(431) In_2_O_3_	8	9	12	11	15
(440) In_2_O_3_	28	50	39	32	53
(433) In_2_O_3_	13	14	7	10	16
(444) In_2_O_3_	15	10	9	9	10

**Table 2 materials-16-00362-t002:** SThM signals and thermal conductivities of reference and ITO samples. The thermal conductivities of ITO samples are apparent thermal conductivities (not corrected for an influence of substrate).

Sample	SThM Signal, a.u.	*k*, W·m^−1^·K^−1^
Reference		
PET	0.904	0.20
Fuses quartz	0.901	1.5
Glassy carbon	0.887	6.3
YAG	0.877	12
Thin films		
S1 (as received)	0.894	3.7 (apparent)
S2 (annealed in air)	0.886	6.6 (apparent)
S3 (annealed in NHM)	0.891	4.6 (apparent)
S4 (annealed in O_2_)	0.892	4.4 (apparent)

**Table 3 materials-16-00362-t003:** Electrical conductivity, carriers’ concentration and mobility of ITO layers.

	*σ*, kS·cm^−1^	*n*, 10^20^ cm^−3^	*μ*, cm^2^·V^−1^·s^−1^
S1	5.06	8.86	35.64
S2	2.79	5.85	29.73
S3	5.53	10.43	33.10
S4	2.11	4.13	36.19
S5	4.76	9.17	31.77

**Table 4 materials-16-00362-t004:** Thermal conductivities, phonon and electron component of ITO layers.

	*k*, W·m^−1^·K^−1^	*k_el_*, W·m^−1^·K^−1^	*k_ph_*, W·m^−1^·K^−1^
S1	5.1	3.7	1.4
S2	10.6	2.0	8.6
S3	6.7	4.0	2.7
S4	6.4	3.7	2.7

## Data Availability

The data presented in this study are available on request from the corresponding author.

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
