# Peer review of "Influence of Post Processing on Thermal Conductivity of ITO Thin Films"

_materials, 2022, doi:10.3390/ma16010362_

Round 1
Reviewer 1 Report
The authors report the thermal conductivity research of ITO thin films annealed at 400 ℃ in different atmosphere (Air, N2 and O2). The basic properties such as crystallography, morphology, thermal and electrical aspects are conducted. This is a worthy subject for the related fields employing ITO thin films, however, considering of the quality requirements by this journal, this current manuscript cannot be accepted.
1. This paper mainly describe the morphology features by AFM measurements, and the roughness is concentrated to evaluate the effects from post annealing. Actually, the surface and cross-section images via SEM seem more direct and vital for the morphology investigation.
2. The authors fail to explain the relationship between the thermal and electronic conductivity distinction in different annealing conditions. The further experimental analysis and discussion for these results are needed.
3. Some wrong words and sentences must be avoided such as line 58 “Similar observations for ITO thin films annealed in range were shown in [12].” and line 62 “the caries mobility “.
Author Response
(x) English language and style are fine/minor spell check required
Reply: We are grateful for the reviewer's suggestion and for careful reading of our manuscript. We checked the manuscript files again and corrected minor typos and any imperfections.
Reply: We are grateful for the reviewer's suggestions. We have revised and substantially expanded the Results and Conclusion sections of the manuscript and improved the results presented in the manuscript, as requested by the reviewer.
- This paper mainly describe the morphology features by AFM measurements, and the roughness is concentrated to evaluate the effects from post annealing. Actually, the surface and cross-section images via SEM seem more direct and vital for the morphology investigation.
Reply: We thank the Reviewer for this insightful comment. We investigated the surface morphology via AFM working in contact-mode and we put it together with detailed XRD data analysis to identify influence of annealing atmosphere on structure of thin films.
- The authors fail to explain the relationship between the thermal and electronic conductivity
distinction in different annealing conditions. The further experimental analysis and discussion for
these results are needed.
Reply: We thank the Reviewer for this insightful comment. We extended the discussion on thermal and electronic conductivity in the following text and added new graphs to the manuscript:
“The electrical conductivity of reference ITO thin film S1 is 5.06 kS·cm-1. Layers annealed in air and O2, (S2 and S4) show the electrical conductivities equal to 2.79 and 2.11 kS·cm-1, respectively. These are the lowest electrical conductivities. The highest thermal conductivity was obtained for NHM annealed sample S3 – 5.53 kS·cm-1. This results from the highest carriers concentration and relatively high carries mobility. The electrical conductivity, carriers concentration and mobility for ITO samples are depicted in Fig. 4b).
The thermal conductivity connected with carriers kel is defined in terms of the electrical conductivity σ by the Wiedemann-Franz law, stating that carrier contribution of the thermal conductivity kel to the electrical conductivity σ of metals is directly proportional to the temperature T:
kel / σ = LT (5)
where L = 2.44‧10-8 WWK-2 is the Lorentz number.
The thermal conductivity carried by phonons was estimated by subtracting kel from the total thermal conductivity of thin films determined from SThM measurements. The results for the thermal conductivity, the phonon and electron components were gathered in the Table 3. The thermal treatment increased the thermal conductivity value for all samples. The thermal conductivity carried by phonons was constant for samples annealed in NHM and O2, what could suggest that the phonon mean free path should be almost constant for two ITO thin films. This observation is corroborated by topography images of sample S3 and S4. The layers are composed of rounded, densely packed grains and the surface roughness is about 1.3 nm. The phonon component of the thermal conductivity of ITO layer annealed in air (S2) is three times larger in comparison to sample 3 and 4, pointing to phonon heat transport in the thin film. Relatively high thermal conductivity can result from effective phonon transport along needle-like grains. The different trend is also reflected in the topography image. The layer consists of densely packed, needle-like grains arranged in sections resembling texturization. The surface roughness equal to 2 nm is also larger for sample S2.
The differences in surface morphology of ITO layers are not fully consistent with parameters determined from XRD spectra analysis. The mean grain size of ITO layers is equal around 63 nm. This observation could suggest that thermal treatment resulted more in changes of grains orientation and less in changes of mean size of grains. The thermal conductivity carried by electrons is higher in comparison to the phonon part for as-received sample and samples annealed in NHM and oxygen atmosphere. The contribution of kel is about 60% of the total thermal conductivity values. “
- Some wrong words and sentences must be avoided such as line 58 “Similar observations for ITO thin films annealed in range were shown in [12].” and line 62 “the caries mobility “.
Reply: We are grateful for the reviewer's suggestion and for careful reading of our manuscript. We checked the manuscript files again and corrected minor typos and any imperfections.

Reviewer 2 Report
Kaźmierczak-Bałata A. et. al., have developed an original research manuscript investigating the morphology, thermal and electrical properties of indium tin oxide (ITO) thin films. To this mean, they have employed AFM, XRD, SThM, and thermal conductivity analysis. However, the significance of this study and its main difference from other similar published literature has not been adequately presented and discussed. Tons of different papers have investigated these factors and no novelty is presented in this research paper.
Moreover, the abstract is too general, and no significant result is introduced to the readers.
The explained characterization data can be found in the spec sheet of the supplier for the ITO thin films.
Overall, I could not find any satisfying noble information that this study added to the body of knowledge.
Author Response
(x) Moderate English changes required
Reply: We are grateful for the reviewer's suggestion and for careful reading of our manuscript. We checked the manuscript files again and corrected minor typos and any imperfections.
Reply: We are grateful for the reviewer's suggestions. We corrected the Abstract in the manuscript.
Abstract: This work presents the influence of post processing on morphology, thermal and electrical properties of indium tin oxide (ITO) thin films annealed at 400 °C in different atmospheres. The commercially available 170 nm thick ITO layers deposited on glass were used as a starting material. The X-ray diffraction measurements revealed polycrystalline structure with dominant signal from (222) plane for all samples. The annealing reduces the intensity of this peak and causes increase of (221) and (440) peaks. Atomic force microscopy images showed that the surface morphology is typical for polycrystalline layers with roughness not exceeding few nm. Annealing in the oxygen and the nitrogen-hydrogen mixture (NHM) changes shapes of grains. The electrical conductivity decreases after annealing except the one of layer annealed in NHM. Thermal conductivities of annealed ITO thin films were in range from 6.4 to 10.6 W·m-1K-1, and they were higher than the one for starting material – 5.1 W·m-1K-1. Present work showed that annealing can be used to modify properties of ITO layers to make them useful for specific applications e.g., in ITO based solar cells.
Reply: We expanded the Introduction and added new references [5, 6, 7, 13, 32]
Investigations of annealing temperature influence in range (150-550) °C on properties of ITO thin films was shown in [16]. The ITO layers have been fabricated by the sol-gel process and post annealed in various temperatures. The crystallinity, optical and electrical properties were improved, what was especially pronounced for temperature equal to 550 °C. Physical properties such as the thermal conductivity, the free carrier density, the carriers mobility, and optical properties are crucial parameters determining application of ITO layers. These parameters are sensitive to microstructure and quality of the layer. The thermal conductivity depends on morphology and thickness of thin film and typically decreases with decrease of layer thickness, what is especially noticeable for thicknesses below 1 mm [17-20]. This fact contributes to difficulties of theoretical predictions of thermal properties of nano-scale layers. Thermal properties of thin films can be modified by doping and by changes of crystalline structure, which depend on fabrication and post processing parameters.
Reply: We have revised and substantially expanded the Results and Conclusion sections of the manuscript and improved the results presented in the manuscript, as requested by the reviewer. We agree with Reviewer, that some of ITO properties (optical, electrical) can be easily found in the literature, however the thermal conductivity of thin films depends on many factors; layer structure, defects, thickness. This parameter is difficult to predict by theoretical analyses, thus it should be verified experimentally for each time after modification of thin film (doping, annealing).
We added following text in the Introduction:
The thermal conductivity depends on morphology and thickness of thin film and typically decreases with decrease of layer thickness, what is especially noticeable for thicknesses below 1 mm [17-20]. This fact contributes to difficulties of theoretical predictions of thermal properties of nano-scale layers. Thermal properties of thin films can be modified by doping and by changes of crystalline structure, which depend on fabrication and post processing parameters. Thus, understanding of thermal transport in thin films is important for engineering thermal properties for selected applications.
Reply: We expanded the Results and Discussion, added new graphs and following text:
The mean crystallite size L was estimated from the Debye-Scherrer’s formula [32]:
, (1)
where k is shape factor, l is the X-rays wavelength, q is the Bragg diffraction angle and b is the broadening of the diffraction peak measured at half of its maximum intensity (FWHM). The shape factor value k = 0.9 was assumed in the calculations. The L parameter varied in range (60 – 66) nm. The analysis of the XRD signal amplitude for all identified peaks shows that, except for one reflection, all signals come from the In2O3 compound. Detailed analysis of spectra revealed that the (222) reflection peak is composed from two components, In2O3 (222) and In4Sn3O12 (300) peaks. (Fig. 1b). The b parameter of In4Sn3O12 peak is the largest among all peaks, which may indicate low crystallinity of this compound in the form of nanocrystallites woven into In2O3 grains.
The amplitudes of all identified reflection peaks corresponding to In2O3 and In4Sn3O12 compounds basing on XRD spectra analysis are gathered in the Table 1. The quantitative analysis of amplitudes of the diffraction peaks shows that annealing in different atmospheres resulted in a decrease of the signal for the dominant (222) reflection, which may indicate a deterioration of crystallinity of the layers. However, in the case of (300) reflection for In4Sn3O12 compound, the signal amplitude increased for all samples except for sample S4 annealed in O2. The amplitude of XRD spectra for ITO layers is depicted in Fig. 1c.
The electrical conductivity of reference ITO thin film S1 is 5.06 kS·cm-1. Layers annealed in air and O2, (S2 and S4) show the electrical conductivities equal to 2.79 and 2.11 kS·cm-1, respectively. These are the lowest electrical conductivities. The highest thermal conductivity was obtained for NHM annealed sample S3 – 5.53 kS·cm-1. This results from the highest carriers concentration and relatively high carries mobility. The electrical conductivity, carriers concentration and mobility for ITO samples are depicted in Fig. 4b).
The thermal conductivity connected with carriers kel is defined in terms of the electrical conductivity σ by the Wiedemann-Franz law, stating that carrier contribution of the thermal conductivity kel to the electrical conductivity σ of metals is directly proportional to the temperature T:
kel / σ = LT (5)
where L = 2.44‧10-8 WWK-2 is the Lorentz number.
The thermal conductivity carried by phonons was estimated by subtracting kel from the total thermal conductivity of thin films determined from SThM measurements. The results for the thermal conductivity, the phonon and electron components were gathered in the Table 3. The thermal treatment increased the thermal conductivity value for all samples. The thermal conductivity carried by phonons was constant for samples annealed in NHM and O2, what could suggest that the phonon mean free path should be almost constant for two ITO thin films. This observation is corroborated by topography images of sample S3 and S4. The layers are composed of rounded, densely packed grains and the surface roughness is about 1.3 nm. The phonon component of the thermal conductivity of ITO layer annealed in air (S2) is three times larger in comparison to sample 3 and 4, pointing to phonon heat transport in the thin film. Relatively high thermal conductivity can result from effective phonon transport along needle-like grains. The different trend is also reflected in the topography image. The layer consists of densely packed, needle-like grains arranged in sections resembling texturization. The surface roughness equal to 2 nm is also larger for sample S2.
The differences in surface morphology of ITO layers are not fully consistent with parameters determined from XRD spectra analysis. The mean grain size of ITO layers is equal around 63 nm. This observation could suggest that thermal treatment resulted more in changes of grains orientation and less in changes of mean size of grains. The thermal conductivity carried by electrons is higher in comparison to the phonon part for as-received sample and samples annealed in NHM and oxygen atmosphere. The contribution of kel is about 60% of the total thermal conductivity values.
We kindly ask the Reviewer to look into our corrected manuscript.

Reviewer 3 Report
Bałata et al. deposited the commercially available ITO on the glass and studied the morphology, thermal, and electrical properties of indium tin oxide (ITO) thin films.
1) The author used the thickness of ITO film to be 170 nm. The authors should provide SEM or TEM images to confirm the thickness.
2) In XRD, the authors should index all the XRD peaks and Rietveld refinement and provide more information about phase and structure.
3) The author should provide XPS information and study the effect of annealing temperature.
4) The value of Ra and grain is changed with annealing temperature and environment. The authors should describe the reason.
5) The authors did not provide information about the rising and cooling temperatures. The authors consider the effect of temperature ramping.
6) Many spelling and formatting typos in this paper, and we hope the authors should check and revise them thoroughly.
Author Response
(x) Extensive editing of English language and style required
Reply: We are grateful for the reviewer's suggestion and for careful reading of our manuscript. We checked the manuscript files again and corrected typos and any imperfections.
- The author used the thickness of ITO film to be 170 nm. The authors should provide SEM or TEM images to confirm the thickness.
Reply: We thank the Reviewer for this insightful comment. We had no possibility to do SEM/TEM images because of the fact that the University is closed in two last weeks of December and beginning of January to save energy. We assumed the thickness provided by the producer. Moreover we did not observe significant changes in morphology of annealed layer, based on careful XRD spectra analysis.
- In XRD, the authors should index all the XRD peaks and Rietveld refinement and provide more information about phase and structure.
Reply: We thank the Reviewer for this insightful comment. We expanded the paragraph connected with XRD analysis, we index all peaks, did Rietveld refinement and provide more information about phase and structure, as requested by the reviewer. We added new graphs and following text:
The structure of ITO was studied by analysis of XRD spectra recorded under conventional (0-2θ) scanning configuration. The ITO thin films show the bixbyite type cubic lattice structure of space group Ia3 (206). The lattice constant a of undoped In2O3 structure is equal to 10.1195 , while substitution of In3+ with Sn4+ ions results in the increase of a to 10.1517 . The characteristic reflections from the following planes: (211), (222), (332), (431), (440), (433), (444) with corresponding peaks positions for In2O3 and In4Sn3O12 components in the XRD pattern are depicted in Fig. 1a) [30, 31]. The reflection from (222) plane corresponding to In2O3 component was dominant for all samples, however the fraction of In4Sn3O12 was also noticeable, as shown in Fig. 1b). The decrease in the intensity of characteristic reflection from plane (222) after annealing, compared to as-received sample, could result from more compacted layers or possible degradation of the film quality after annealing at 400 °C in air, NHM, O2 and N2 atmosphere.
The mean crystallite size L was estimated from the Debye-Scherrer’s formula [32]:
, (1)
where k is shape factor, l is the X-rays wavelength, q is the Bragg diffraction angle and b is the broadening of the diffraction peak measured at half of its maximum intensity (FWHM). The shape factor value k = 0.9 was assumed in the calculations. The L parameter varied in range (60 – 66) nm. The analysis of the XRD signal amplitude for all identified peaks shows that, except for one reflection, all signals come from the In2O3 compound. Detailed analysis of spectra revealed that the (222) reflection peak is composed from two components, In2O3 (222) and In4Sn3O12 (300) peaks. (Fig. 1b). The b parameter of In4Sn3O12 peak is the largest among all peaks, which may indicate low crystallinity of this compound in the form of nanocrystallites woven into In2O3 grains.
The amplitudes of all identified reflection peaks corresponding to In2O3 and In4Sn3O12 compounds basing on XRD spectra analysis are gathered in the Table 1. The quantitative analysis of amplitudes of the diffraction peaks shows that annealing in different atmospheres resulted in a decrease of the signal for the dominant (222) reflection, which may indicate a deterioration of crystallinity of the layers. However, in the case of (300) reflection for In4Sn3O12 compound, the signal amplitude increased for all samples except for sample S4 annealed in O2. The amplitude of XRD spectra for ITO layers is depicted in Fig. 1c.
Table 1. Amplitudes of pronounced reflection peaks corresponding to In2O3 and In4Sn3O12 compounds.
|
|
S1 |
S2 |
S3 |
S4 |
S5 |
|
(221) In2O3 |
29 |
40 |
63 |
47 |
50 |
|
(300) In4Sn3O12 |
263 |
283 |
322 |
205 |
320 |
|
(222) In2O3 |
763 |
565 |
517 |
606 |
658 |
|
(332) In2O3 |
44 |
39 |
23 |
31 |
43 |
|
(431) In2O3 |
8 |
9 |
12 |
11 |
15 |
|
(440) In2O3 |
28 |
50 |
39 |
32 |
53 |
|
(433) In2O3 |
13 |
14 |
7 |
10 |
16 |
|
(444) In2O3 |
15 |
10 |
9 |
9 |
10 |
Figure 1. XRD spectra recorded for ITO thin films: as-received (REF) and annealed at 400 °C in various atmospheres (a), exemplary diffraction peak analysis from plane (222) for as-received sample (b), amplitudes of reflection signals for ITO layers c).
- The author should provide XPS information and study the effect of annealing temperature.
Reply: We are able to do XPS, the experiment can be done in January 2023, because the University is closed to 8th January 2023. However the XPS provides information about chemical composition only from subsurface region, the information comes from surface of the layer. The XRD measurements provide information about whole crystal structure from the volume of the layer.
- The value of Ra and grain is changed with annealing temperature and environment. The authors should describe the reason.
Reply: We are grateful for the reviewer's suggestion, we added the following text in the paragraph 3.3:
“This observation is corroborated by topography images of sample S3 and S4. The layers are composed of rounded, densely packed grains and the surface roughness is about 1.3 nm. The phonon component of the thermal conductivity of ITO layer annealed in air (S2) is three times larger in comparison to sample 3 and 4, pointing to phonon heat transport in the thin film. Relatively high thermal conductivity can result from effective phonon transport along needle-like grains. The different trend is also reflected in the topography image. The layer consists of densely packed, needle-like grains arranged in sections resembling texturization.
The surface roughness equal to 2 nm is also larger for sample S2. The differences in surface morphology of ITO layers are not fully consistent with parameters determined from XRD spectra analysis. The mean grain size of ITO layers is equal around 63 nm. This observation could suggest that thermal treatment resulted more in changes of grains orientation and less in changes of mean size of grains.”
- The authors did not provide information about the rising and cooling temperatures. The authors consider the effect of temperature ramping.
Reply: We are grateful for the reviewer's suggestion, the information is already in the paragraph 2:
The heating and cooling rates were about 3 °C/min.
6) Many spelling and formatting typos in this paper, and we hope the authors should check and revise them thoroughly.
Reply: We are grateful for the reviewer's suggestion and for careful reading of our manuscript. We checked the manuscript files again and corrected minor typos and any imperfections.

Reviewer 4 Report
There are some adjustments to do as using: /value/ {unit}. Some of them are signed in the pdf notes.
There is a point that should be clearly stated to justify completely your discussion of the results: On figure 3 we have an idea of the SThM signal error but the thermal conductivity error is not given. The apparent thermal conductivity scales with the measured thermal conductivity, but the absolute values are very different.
The correlation that you made for the references is not physically justified? There is a calibration problem? Or the error in measured values of k are significant?

Author Response
(x) English language and style are fine/minor spell check required
Reply: We are grateful for the reviewer's suggestion and for careful reading of our manuscript. We checked the manuscript files again and corrected minor typos and any imperfections.
There are some adjustments to do as using: /value/ {unit}. Some of them are signed in the pdf notes.
Reply: We are grateful for the reviewer's suggestions. We have revised our manuscript according pdf file.
There is a point that should be clearly stated to justify completely your discussion of the results: On figure 3 we have an idea of the SThM signal error but the thermal conductivity error is not given. The apparent thermal conductivity scales with the measured thermal conductivity, but the absolute values are very different.
The correlation that you made for the references is not physically justified? There is a calibration problem? Or the error in measured values of k are significant?
Reply: We are grateful for the reviewer's comment. The SThM signal uncertainty is really small, of order 1-2% calculated as a standard uncertainty based on the number of repeated measurements, thanks to applying lock-in detection. However, the thermal conductivity uncertainties are significant, as suggested by the Reviewer, and can reach +/- 1.3 Wm-1K-1, what is 20-25% of k value.
The thermal conductivity uncertainties are given in the text, they are in the range 15-20%. (line 209)
We have also revised and substantially expanded the Results and Conclusion sections of the manuscript and improved the results presented in the manuscript, as requested by the reviewer.
We kindly ask the Reviewer to look into our corrected manuscript.

Round 2
Reviewer 1 Report
The manuscript is improved and can be accepted in current version.
Reviewer 3 Report
Accept in present form